# OpenReview forum: "Siege: Multi-Turn Jailbreaking of Large Language Models with Tree Search"
_ICLR.cc/2025/Workshop/BuildingTrust — BuildingTrust_

### Official Review · Reviewer_bMGo · 2025-02-27
**TEMPEST: Multi-Turn Jailbreaking of Large Language Models with Tree Search**

**Rating:** 7
**Confidence:** 3

**Review:**

This paper presents a compelling examination of a critical flaw in AI safety—while many models perform well on single-turn safety tests, they often fail under multi-turn adversarial conditions. Through rigorous experimentation, the study demonstrates that its proposed approach is significantly more effective than prior methods, necessitating a reassessment of existing AI defense strategies. By highlighting the insufficiency of current safety measures, the paper makes a strong case for incorporating multi-turn adversarial testing into standard evaluation frameworks. Moreover, while the study primarily focuses on exploiting AI vulnerabilities, its methodology could also be leveraged to enhance AI security by enabling models to “remember” previous interactions rather than treating each prompt in isolation. Additionally, further evaluation on diverse datasets beyond JailbreakBench, including real-world adversarial prompts, could strengthen the findings’ generalizability. Finally, the partial compliance metric introduced in the paper is a valuable contribution, but a more detailed explanation of how intermediate compliance levels (1–9) are assigned—beyond the clear-cut cases of full refusal (0) and complete compliance (10)—would enhance clarity and reproducibility.

---

### Official Review · Reviewer_63A4 · 2025-03-01
**Review of "TEMPEST: Multi-Turn Jailbreaking of Large Language Models with Tree Search"**

**Rating:** 7
**Confidence:** 4

**Review:**

The paper introduces a method for jailbreaking LLMs (i.e., making them generate potentially harmful outputs that have been discouraged during the alignment process). The approach is based on multi-turn conversations, where each turn is encoded as a node in a tree. It employs breadth-first tree search to obtain the requested, though forbidden, response. Nodes that elicit partial compliance from the model are considered promising and are used as starting points for the next conversation turn. The method outperforms other jailbreaking baselines while requiring fewer queries to the model.

**Positive Aspects:**
- Effective, intuitive, and reasonably simple method for jailbreaking LLMs.
- Clear explanation of the method.
- Comparison to several baselines, with the proposed method demonstrating clear superiority.

**Negative Aspects:**
- The paper lacks sufficient experimental details to ensure reproducibility. For example, what are the exact prompts used for the attacker LLM?
- The partial compliance function $\gamma$ appears to be a crucial component of the method, but the paper does not explain how compliance is measured in practice.

**Additional Comment:**
- Why is the method called "TEMPEST"? The capitalization suggests that it is an acronym, but this is never explained. Simply naming a method for the sake of branding feels like a marketing trick, which, in my view, is not appropriate for a research paper.

---

### Decision · Program_Chairs · 2025-03-04

Accept